# Changes in the Mechanical Properties of Fast and Slow Skeletal Muscle after 7 and 21 Days of Restricted Activity in Rats

**DOI:** 10.3390/ijms24044141

**Published:** 2023-02-18

**Authors:** Sergey A. Tyganov, Svetlana P. Belova, Olga V. Turtikova, Ivan M. Vikhlyantsev, Tatiana L. Nemirovskaya, Boris S. Shenkman

**Affiliations:** 1Myology Laboratory, Institute of Biomedical Problems RAS, 123007 Moscow, Russia; 2Laboratory of Structure and Functions of Muscle Proteins, Institute of Theoretical and Experimental Biophysics, Russian Academy of Sciences, 142290 Pushchino, Russia

**Keywords:** soleus muscle, extensor digitorum longus, movement restriction, decreased muscle activity, cytoskeletal proteins, mechanical properties

## Abstract

Disuse muscle atrophy is usually accompanied by changes in skeletal muscle structure, signaling, and contractile potential. Different models of muscle unloading can provide valuable information, but the protocols of experiments with complete immobilization are not physiologically representative of a sedentary lifestyle, which is highly prevalent among humans now. In the current study, we investigated the potential effects of restricted activity on the mechanical characteristics of rat postural (soleus) and locomotor (extensor digitorum longus, EDL) muscles. The restricted-activity rats were kept in small Plexiglas cages (17.0 × 9.6 × 13.0 cm) for 7 and 21 days. After this, soleus and EDL muscles were collected for ex vivo mechanical measurements and biochemical analysis. We demonstrated that while a 21-day movement restriction affected the weight of both muscles, in soleus muscle we observed a greater decrease. The maximum isometric force and passive tension in both muscles also significantly changed after 21 days of movement restriction, along with a decrease in the level of collagen 1 and 3 mRNA expression. Furthermore, the collagen content itself changed only in soleus after 7 and 21 days of movement restriction. With regard to cytoskeletal proteins, in our experiment we observed a significant decrease in telethonin in soleus, and a similar decrease in desmin and telethonin in EDL. We also observed a shift towards fast-type myosin heavy chain expression in soleus, but not in EDL. In summary, in this study we showed that movement restriction leads to profound specific changes in the mechanical properties of fast and slow skeletal muscles. Future studies may include evaluation of signaling mechanisms regulating the synthesis, degradation, and mRNA expression of the extracellular matrix and scaffold proteins of myofibers.

## 1. Introduction

A prolonged reduction in skeletal muscle activity, such as during confinement to bed or due to a sedentary lifestyle, is highly prevalent in humans. Moreover, the recent pandemic situation forced people to reduce their activity and adopt sedentary behaviors. The use of various models of reduced muscle activity have shown that disuse induces profound muscle loss and has a significant impact on the mechanical and morphological properties of muscles [1,2,3]. The dry immersion model, bed rest, limb immobilization, and other models can provide valuable information on the detrimental effects of inactivity, but the translation of these extreme experimental models into less severe forms of inactivity in free-living environments is difficult [4]. Thus, models of prolonged isolation [5] and step reduction [6,7,8] have been used to study decreased activity in human experiments, and the model of restriction of mobility in small cages has been used in animal experiments [9,10,11]. There are comparably fewer studies exploring the influence of such physical inactivity on changes in the mechanical characteristics of skeletal muscles [12].

Skeletal muscles consist of “slow-type” (high fatigue resistance, low speed of relaxation/contraction, low force) and “fast type” (low fatigue resistance, high speed of relaxation/contraction, high force) fibers. The predominance of one of these fiber types in a muscle determines its phenotype. Skeletal muscle disuse affects postural and locomotor muscles in different ways. Various disuse models (spaceflight, hindlimb suspension, spinal cord isolation, etc.) lead to atrophy of both muscle types, but the degree and kinetics of this atrophy are very different (usually more pronounced in muscle fibers expressing an abundance of the slow-type myosin heavy chains, MyHCs) [13]. Disuse muscle atrophy is accompanied by changes in the myosin phenotype [14,15,16], contractile properties [17,18], and elastic properties [18,19,20,21]. However, there are few data on the effect of reduced activity on mechanical parameters and skeletal muscle composition. Young adults exhibit a decrease in maximal aerobic capacity, voluntary strength of the lower limbs, and knee extensor maximum contraction strength in response to step reduction of varying duration [8]. One week of step reduction by 91% lowered daily myofibrillar protein synthesis by 27% [7]. In a human prolonged confinement experiment, it was shown that countermovement jump power and single-leg hop force decreased during isolation [5], and a significant loss of maximal voluntary isokinetic force in the quadriceps/hamstrings has also been shown [22]. In another study, it was shown that step reduction from an average ~14,000 steps/d to ~3000 steps/d induced a decline in MyHCI (myosin heavy chain) and MyHCII muscle fiber cross-section area (CSA) on the mid vastus lateralis in healthy, trained men [23]. Two weeks of reduced daily activity (1413 ± 110 steps per day) induced a decrease in leg fat-free mass and muscle protein synthesis in the healthy elderly [24]. In animal experiments, it has been shown that movement restriction for 14 and 28 days decreased soleus muscle mass, grip force and cross-sectional area [10]. Housing mice in small cages for 19 weeks led to reduced motor coordination, grip strength and muscle stamina [25], and other authors have shown that housing mice in small cages for 4 weeks decreased tibialis anterior muscle mass, the CSA of myofibers and grip strength [26]. Recently, it was also shown that prolonged physical inactivity (8 weeks) exacerbates hindlimb unloading-induced disuse muscle atrophy in rat soleus, but this model itself did not affect the skeletal muscle [11]. In the study of Takemura et al., it was hypothesized that the restricted activity model would preserve the load on the postural muscles but significantly reduce the activity of the locomotor muscles. Therefore, fast locomotor muscles should change the most after movement restriction [9].

The purpose of the present study was to determine the effects of rat movement restriction on the mechanical properties and cytoskeleton composition of postural (soleus) and locomotor muscle EDL. We hypothesized that the effects of decreased activity would be different from those of full mechanical unloading when there is no influence from support and body weight (e.g., space flight, dry immersion, bedrest, etc.). We also hypothesized that changes in response to decreased muscular activity will differ in “fast” and “slow” muscle.

## 2. Results

### 2.1. Muscle Weight and Mechanical Characteristics

The body weight of the animals did not differ significantly between the groups. Soleus and EDL mass significantly decreased only in the R21 group by 11% and 6%, respectively. 

Restriction of activity (R7 and R21) did not lead to a change in the twitch contraction parameters (maximum strength, time of contraction, and half-relaxation time) of soleus. However, we observed a decrease in the maximum force of tetanic contraction in the R21 group relative to the control group by 34% (Figure 1A). For the specific maximum force of tetanic contraction (i.e., the force divided by the cross-sectional area of the muscle), we observed a decrease in the R21 group by 30%. The passive mechanical characteristics were also measured using fast (50 mm/s) 25% stretch of slack length without electrical field stimulation. We observed an increase in the maximum peak tension (Fp) (peak in the beginning of the stretch) by 29% and in the specific maximum peak stiffness (Fp/CSA) by 44% in soleus in the R21 group (Figure 1A). In addition, we observed an increase in Young’s modulus E1 (elastic element 1, or a stress/strain ratio for the first elastic element in the viscoelastic model of skeletal muscle) and E2 (elastic element 2) in the R21 group by 82% and 50%, respectively, in soleus.

In EDL, we observed a significant decrease in the absolute and specific maximum twitch tension in the R21 group (Figure 1B). The maximum and specific maximum tension of tetanic contraction also significantly decreased by 55% and 54%, respectively, in R21 relative to the control group. For EDL, we observed a significant decrease in both maximum peak tension (Fp) and steady tension (Fs) by 41% and 32%, respectively (Figure 1B). Likewise, restricted activity for 21 days reduced the stiffness and Young’s modulus E1 and E2 of EDL. Data on all mechanical measurements of both muscles are presented in the Appendix A.

### 2.2. Collagen Content and mRNA Expression

To assess a possible contribution of the ECM to the passive mechanical characteristics of rat soleus, we determined the collagen content and the expression levels of collagen 1a and 3a mRNA. After both 7-day and 21-day restriction of activity, we observed a decrease in collagen content in soleus, but not in EDL (Figure 1C,D). We did not observe any changes in collagen 1 mRNA content in soleus and EDL (Figure 1E,F); however, collagen 3 mRNA expression significantly decreased in the R7 and R21 groups by 22% and 23%, respectively, in soleus (Figure 1E). In EDL, collagen 3 mRNA content was also significantly decreased in the R7 and R21 groups by 32% and 23%, respectively (Figure 1F).

### 2.3. MyHC mRNA Transcription

The transcription of slow MyHC I mRNA in soleus in the R7 group significantly decreased compared with the control group by 22%; however, we did not observe any changes of MyHC I mRNA content in R21 group. Fast MyHC IIa, IIb and IId/x mRNA transcription significantly increased in R21 group. We also observed a significant increase in MyHC IId/x mRNA in R7 group (Figure 2A). In EDL, we did not observe any changes in MyHC mRNA content in the R7 or R21 groups (Figure 2B), nor did we observe any shift in MyHC ratio following immunohistochemical analysis of soleus and EDL in the R21 group (Figure 3A,B). We also did not observe any significant changes in fast and slow muscle fiber CSA (Figure 3C,D)

### 2.4. Cytoskeleton Protein Content and mRNA Transcription

To assess the contribution of cytoskeletal proteins to any unloading-induced reduction in soleus mechanical properties, Western blot analysis and PCR were carried out. Desmin (muscle-specific intermediate filament protein) mRNA expression significantly decreased after 7-day restriction of activity in soleus, and after 21-day restriction in EDL (Figure 4A). Desmin protein content was unchanged in soleus after 7 and 21 days of restriction (Figure 5A,B). In EDL, desmin protein content was significantly reduced in the R21 group (Figure 5C,D) with no changes in desmin mRNA (Figure 4B). Actinin-2 (which anchors myofibrillar actin and titin to the Z-disc and is specific to oxidative myofibers) mRNA expression did not change in soleus and was significantly increased in EDL in the R7 group (Figure 4A,B). Actinin-2 protein content was unchanged both in soleus and EDL in the R7 and R21 groups (Figure 5B,D). Actinin-3 (which is specific to glycolytic myofibers) mRNA significantly increased in soleus in the R21 group and decreased in EDL in the R7 group (Figure 4A,B). Actinin-3 protein content was unchanged both in soleus and EDL in the R7 and R21 groups (Figure 5A–D). We did not observe any changes in telethonin (which connects titin with the Z-disc) mRNA expression in either muscle (Figure 4A,B). However, telethonin protein content significantly decreased in soleus in the R7 group and in EDL in the R21 group (Figure 5A–D).

We also analyzed titin (intact titin (T1) and proteolytic fragment (T2)) and nebulin protein content. We did not observe any changes in giant protein content after 21-day movement restriction (Figure 6A–C).

## 3. Discussion

In the current study, we investigated the effects of 7 and 21 days of restricted activity on the mechanical characteristics of rat soleus and EDL. We demonstrated that 21-day movement restriction affected the weight of both muscles, but we observed a greater decrease in the weight of soleus. The mechanical characteristics of both muscles also significantly altered after 21 days of movement restriction, along with a decrease in collagen 1 and 3 mRNA expression levels. At the same time, the collagen content itself changed only in soleus after 7 and 21 days of movement restriction. Regarding the cytoskeletal proteins, we observed a significant decrease in telethonin protein content in soleus, and a similar decrease in desmin and telethonin content in EDL.

The models of movement restriction have led to different levels of muscle mass decrease depending on the extent of the activity restriction, due to the different sizes and structures of the cages used. In the experiments of Takemura et al. the activity of animals was reduced by 200 times (27 cm*d^−1^ and 5482 cm*d^−1^), and the duration of the experiment was 21 days [9]. A 21-day movement restriction led to a non-significant decrease in the weight of soleus by 10% and a significant decrease in the weight of plantaris (a soleus agonist, but significantly “faster” in phenotype) by 27% [9]. In a study performed by Marmonti et al., the activity of the animals was not measured, and the duration of the experiment was 7, 14, and 28 days. In that work, a decrease in the mass of soleus by 7.3, 10.5, and 13.2% was observed at 7, 14, and 28 days of the experiment, respectively. At the same time, movement restriction did not lead to changes in the weight of gastrocnemius, EDL, and tibialis anterior [10]. In recent experiments by Yoshihara et al., the third model of mobility restriction was investigated and showed a 10-fold decrease in activity within 21 days, which did not lead to a decrease in the mass of soleus [11]. A 28-day restriction also caused a decrease in the mass of tibialis anterior [26]. In another experiment, a decrease in the weight of soleus was shown in mice kept in small cages, in which the ability to lid climb was permitted [25]. In our study, we observed a small but significant decrease in soleus and EDL mass after 21 days of movement restriction. From the above data, it is noticeable that the response of postural and locomotor muscles to reduced activity with small cages was very different and depended on the cage design and duration of exposure. It is possible to compare the different restricted mobility studies on the basis of the size of the cages. The largest cell size reduction of total space (by 80%) was seen in the work of Marmonti et al. (cage size 12 × 8 × 12 cm), in which the greatest amount of soleus atrophy was observed [10]. In the experiment of Takemura et al., the size of the cells was decreased in both the width and the length by 50% (cage size 17 × 9.6 × 13 cm), and in this work atrophy was observed only for plantaris [9]. Finally, in the experiments of Roemers et al. and Yoshihara et al., where only the size of the floor space was reduced by 50% (cage size 16.5 × 13 × 13 cm and 18 × 11 × 11 cm, respectively), atrophy was observed for different muscles (for the soleus in the work of Roemers et al. and for the tibialis anterior in the work of Yoshihara et al.) [11,25]. The data in the current study (cage size 17 × 9.6 × 13, 50% reduction in cell length and height) indicated significant, though small, changes in the weight of soleus and EDL after 21 days of restriction, with more pronounced changes in soleus weight. Thus, it can be seen that the atrophic processes that occur in skeletal muscles in the model of movement restriction are apparently not only related to the cell size. We believe that in our study, the locomotor activity of animals inside the cells played a significant role, which for some reason differed in all the aforementioned works. The main function of the soleus is considered to be resistance to gravitational forces, thus maintaining the animal’s vertical posture. However, as evidenced by electromyographic studies under conditions of free behavior of the animal, the electromyography activity of soleus associated with locomotion is as large in amplitude as its postural electromyography activity [27]. Therefore, restricting the locomotor activity of the animal could also affect soleus mass, as happened during gravitational unloading using the hindlimb suspension model.

Changes in the mechanical characteristics of the calf muscles after movement restriction have not been studied thoroughly. It has been shown that movement restriction for 7, 21, and 28 days leads to a significant decrease in the grip strength of the hindlimbs of rats [10,26]. The all-limbs grip strength and hindlimbs grip strength significantly decreased in another experiment on mice, which were limited in mobility in small cages and were prevented from lid climbing for 19 weeks [25]. In one of the experiments with small cages, the mechanical characteristics of isolated mouse muscles were also studied. A 6-week movement restriction did not cause changes in either the passive or active properties of soleus and tibialis anterior, with no change in the weight of these muscles [28]. In the current study, we observed a decrease in the maximum force of tetanic contraction of isolated soleus and EDL only after 21 days of movement restriction, which generally correlates with previously obtained data on the grip strength of animals in experiments with restriction. However, for EDL we observed a decrease in passive stiffness, in contrast with soleus, which showed an increase in both peak and stable passive characteristics. This phenomenon had not previously been seen in a mobility restriction model, and to study it, we analyzed different types of myosin heavy chains, cytoskeletal proteins, and extracellular matrices. When analyzing the changes in the active and passive mechanical properties of soleus and EDL with restricted activity, the differences between this model and the model of disuse should be taken into account. In contrast with the disuse model, in the restriction model all structures and mechanisms that contribute to maintaining a vertical posture against gravitational forces are expected to remain active despite a decrease in the volume of locomotion. Therefore, it is difficult to imagine that in the restriction model, a decrease in the stiffness characteristics of soleus—the main anti-gravity muscle of mammals—would be observed. At the same time, it is possible that in normal soleus activity, the dynamic and static properties are in balance and, possibly, in a reciprocal relationship. In this case, with a decrease in the involvement of dynamic properties, the stiffness characteristics of soleus may even increase, which we observed after a 21-day restriction of locomotor activity. Interestingly, this increase is accompanied by a decrease in the expression of collagen III mRNA. It is known that collagen III, which is more characteristic of fast muscles, exhibits greater elasticity and less rigidity than collagen I, which usually accompanies slow type I fibers [29,30].

“Fast” type fibers have a relatively greater strength and speed of contraction compared with “slow” type fibers [13,31]. Therefore, the changes in the active mechanical properties of soleus and EDL after restriction could be associated with the observed changes in myosin mRNA content, which determines the active muscle characteristics of the fiber. Changes in the expression of mRNA of different types of myosin heavy chains have not previously been shown in movement restriction experiments, but this shift has been well studied for hindlimb suspension. Numerous experiments have shown that suspension leads to a shift in mRNA expression from MyHCI and MyHCIa to MyHCIIb and MyHCIId/x [13,32,33,34,35]. In one study, a progressive increase in the protein content of MyHCIIa, MyHCIIb, MyHCIId/x and a decrease in the content of MHCI were shown at 7, 15, and 21 days of suspension [32]. At the same time, the shift in the myosin content to the “fast” phenotype after 7 days of suspension corresponded to a decrease in the active and passive mechanical characteristics of soleus [36]. The content of myosin heavy chains of different types was also studied using the immunohistochemical method in an experiment with movement restriction, in which, the authors did not observe a shift in the myosin phenotype [9]. In the current experiment, we observed a noticeable shift in myosin mRNA expression towards the “fast” phenotype after 21 days of movement restriction in soleus, but not in EDL. In addition, we observed a decrease in the maximum contraction force of both muscles, as well as multidirectional changes in passive stiffness characteristics. These changes clearly disagree with the data on the expression of myosin heavy chains, which means that changes in the sarcomeric and non-sarcomeric cytoskeleton of the muscle fiber may affect the mechanical properties of soleus and EDL.

The distance between the actin and myosin filaments must be optimal for the best cross-bridge formation. This condition is provided by a number of cytoskeletal proteins that maintain myofilaments in the desired position and coordinate the work of actin and myosin filaments [37]. The elimination of even one component of the cytoskeletal network can lead to the destruction of the entire sarcomere scaffold, which is shown in models with a deficiency of dystrophin [38], desmin [39], α-actinin-2 [40], or telethonin [41]. Previously, changes in the content and expression of mRNA of various cytoskeletal proteins were studied at different periods of hindlimb suspension. It has been shown that the content of cytoskeletal proteins decreases in soleus after 6 weeks of suspension, and dystrophin complex protein content decreases as early as 3 weeks [42]. These changes were missing in EDL. However, in EDL, at 3 and 6 weeks of unloading, the content of the components of the dystrophin–glycoprotein complex (costamere component) increased. It is noteworthy that the content of desmin and α-actinins did not change in this work [42]. In our recent experiments on 7-day HS, a decrease in the content of α-actinin-2, α-actinin-3, desmin, titin, and nebulin was shown [36]. The same work showed a decrease in the mRNA content of α-actinin-2, desmin, and titin [36]. In the current experiment, we observed a decrease in telethonin in soleus on the seventh day of movement restriction, and in EDL on the twenty-first day. Telethonin is a protein that provides a tight bond between two anti-parallel strands of titin in the Z-disc [37]. In telethonin knockout experiments, it was shown that the absence of this protein did not lead to changes in maximal contraction force or fatigue in isolated soleus and EDL. However, telethonin gene KO led to a change in the stiffness of these muscles [43]. This means that the decrease in the passive stiffness of EDL on the twenty-first day of movement restriction could be associated with a decrease in telethonin. The decrease in the active properties of EDL is also presumably associated with a decrease in desmin (which stabilizes Z-discs) mRNA and protein content. Lack of desmin leads to irregular organization, decreased strength, and increased fatigue in myofibers [44]. It is also interesting that in the current experiment we observed changes in the expression pattern of α-actinin-3 mRNA in soleus. Actinin-3 is expressed exclusively in type II myofibers [37], and an increase in the mRNA of this protein corresponds to the shifting of myosins to the “fast” phenotype, which we observed in soleus.

Recent studies indicate a key contribution of the extracellular matrix to the mechanical properties of muscle fibers [45,46]. Several collagen isoforms are expressed in skeletal muscle, but I and III collagens make the greatest contribution to the mechanical properties of muscle [29]. Moreover, collagen III is characterized by less rigidity and greater elasticity compared with collagen I [20]. The collagen I mRNA content in soleus was analyzed in hindlimb suspension experiments and a pronounced decrease in expression was found on the third day, which was replaced by recovery to the control level after 7 days of suspension [47]. In turn, the expression of collagen III mRNA in soleus decreased after 7 days of suspension [36]. In the “fast” muscle (gastrocnemius), the expression of collagen isoforms also decreased after 3 weeks of suspension [48]. In another work, it was shown that hindlimb suspension for 14 and 28 days led to a shift in the collagen phenotype (from type I to type III) in soleus, but not in the “fast” plantaris [30]. Immobilization experiments have also shown that mechanical unloading results in a decrease in hydroxyproline (an amino acid that provides collagen fiber crosslinks) in the extracellular matrix [49,50]. In the current study, we observed a decrease in the content of collagen 3 mRNA in both soleus and EDL after both 7 and 21 days of movement restriction. Furthermore, histochemical staining for collagen in myofibers showed a decrease in collagen content only in soleus after 7 and 21 days of movement restriction. Thus, in soleus, we observed more profound changes in the composition of the extracellular matrix in comparison with EDL.

## 4. Materials and Methods

### 4.1. Experimental Design

Wistar male rats were obtained from the certified Nursery for laboratory animals of the Institute of Bioorganic Chemistry of the Russian Academy of Sciences (Pushchino, Moscow region, Russia). Three-month-old rats were divided into four groups (*n* = 8): vivarium control for 7 and 21 days (C7 and C21); 7-day and 21-day restriction of activity (R7 and R21). The control animals were kept in normal cages (30.0 × 40.0 × 34.0 cm). The restricted activity rats were kept in small Plexiglas cages (17.0 × 9.6 × 13.0 cm). The cages were arranged in such a way that 4 rats were located next to each other (Figure 7A,B). The animals were kept in a temperature-controlled and light-controlled room on a 12:12-h light-dark cycle. All rats had access to a standard diet with food pellets and water ad libitum. Prior to muscle dissection, the animals were anesthetized with an intraperitoneal injection of tribromoethanol (400 mg/kg). The animals were euthanized by decapitation under deep anesthesia.

### 4.2. Ex Vivo Muscle Mechanical Properties

Ex vivo force measurements of EDL and soleus were assessed as previously described [36]. The isolated muscle optimal length was estimated with digital calipers in situ by placing the knee and ankle joints at a 90° angle. Then each muscle was dissected and placed in a cooled Ringer–Krebs solution (138 mM NaCl, 5 mM KCl, 1 mM NaH_2_PO_4_, 2 mM CaCl_2_, 2 mM MgCl_2_, 24 mM NaHCO_3_, and 11 mM glucose) with constant perfusion with carbogen (95% O_2_ + 5% CO_2_) and incubated for 15 min. Double knots were tied around the distal and proximal ends of the muscle near the musculotendinous junction. After that, the muscle was attached to the lever arm/force transducer from one end and to the fixed hook at the other end in a temperature-controlled (28 °C) water bath (Aurora Scientific Bath 809C). Optimal muscle length (L_0_) was redetermined with a series of twitch contractions induced by electric field stimulation using two parallel electrodes (supra-maximal square wave pulse 0.5 ms, 20 V). The maximum force of twitch contractions was measured at the L_0_. For each twitch contraction the time to peak (time taken to reach maximum force) and half-relaxation time (time taken for force to fall to half of the maximum) was measured. A tetanic isometric contraction test was performed at L_0_. The soleus was stimulated for 2 s at 40 Hz pulse frequency and the EDL was stimulated for 1 s at 120 Hz pulse frequency with two parallel platinum electrodes. The maximum force of tetanic contraction was recorded.

The passive properties of the muscles were also measured on isolated muscle. The soleus and EDL were set to the slack length (Ls) or the length from which the beginning of tension development was measurable. Then the muscle was stretched by 25% of Ls at a speed of 50 mm/s. The length of the muscle was held at 25% of Ls for two minutes, after which the length was returned to Ls [51]. The maximum force was recorded at the beginning (peak tension, Fp) and at the end of the stretch (steady tension, Fs). Passive tension gradually decreased (tension relaxation) and reached a plateau by the second minute of the test (Figure 8).

As it is understood that two elastic elements connected in parallel through a viscous element can serve as a skeletal muscle model [52], Young’s modulus (E1 and E2) was calculated. Young’s modulus is a property of the material that describes how easily it can deform. It was calculated for each element as a stress/strain ratio [53], where stress is the amount of force applied per unit area (σ = F/CSA) and strain is the extension per unit length (25% of L_s_). For all calculations, was used the tension obtained as a result of 5 repetitions for each muscle. To normalize the parameters, the physiological cross-sectional area (CSA) of the muscle was calculated as the muscle wet weight divided by the product of muscle optimal length and density [54]. Force measurements were performed by using an Aurora Scientific Dual Mode Lever System 305C-LR (Aurora, ON, Canada), with a data acquisition frequency of 10 kHz. Data processing was carried out using the Aurora Scientific 615A Analysis Software Suite (Program Version: 5.200, Aurora, ON, Canada).

### 4.3. Collagen Assessment

Collagen histochemical detection was performed in accordance with the method of Segnani C. et al., 2015 [55]. The transverse frozen sections (10 μm thick) of the soleus and EDL muscle samples were prepared with a Leica CM 1900 cryostat (Leica, Braunschweig, Germany) at 20 °C. After that, the sections were fixed in 4% PFA for 20 min and washed in distillate water. The sections were then incubated in 0.04% Light Green (Light green SF yellowish (C.I. 42095), SERVA FEINBIOCHEMICA GmbH & Co. KG, Heidelberg, Germany) for 15 min at room temperature, washed in distillate water for 10 min and incubated in 0.1% Light Green with 0.04% Sirius Red (Direct Red 80, Sigma Aldrich, St. Louis, MO, USA) in saturated picric acid for 30 min. After that, the sections were washed in 0.5% hydrochloric acid and photographed. The sections were analyzed using a Leica Q500MC fluorescence microscope with a built-in digital camera (TCM 300F, Leica, Germany) and objective magnification 400×. Computer analysis of the images was carried out using a custom plugin for ImageJ 1.52a. The total area occupied by collagen on the sections was calculated and expressed in relative units. For detailed images of collagen stained cryosections, please see Appendix A.

### 4.4. Nucleic Acids Analysis

The total RNA fraction was isolated and used for reverse transcription followed by PCR reaction to analyze the expression of mRNA in muscle tissue, using the RNeasy micro kit (Qiagen, Germantown, MD, USA). The following components were used for reverse transcription («Syntol», Moscow, Russia): 30 μM random hexanucleotide mix, 17.4 μM oligo-d(T)15, 1.3 mM dNTP, 0.02 un./μL RNAse inhibitor, 6 un./μL M-MLV-revertase, and 5x-buffer for M-MLV-revertase. The primers used for real-time PCR are shown in Table 1.

### 4.5. Titin and Nebulin Electrophoresis

Titin and nebulin content were assessed as previously described [36]. Giant protein content was analyzed using the technique of SDS-electrophoresis in 2.2% polyacrylamide gel with 0.5–0.6% agarose [56], in order to maximize the preservation of high-molecular isoforms of titin from degradation [57]. To ensure equal loading, samples from the control and experimental groups were all run on the same gel. SDS-PAGE was performed using the Helicon VE–10 system (Moscow, Russia) at 8 mA. Following SDS-PAGE, the gels were stained with Coomassie Brilliant Blue (G-250 and R-250, 1:1). Titin and nebulin content were normalized to the content of MyHC.

### 4.6. MyHC Immunostaining

MyHC determination was carried out as was previously described [58]. Briefly, cross-sections from the muscles were cut at 10 µm in a Leica Microsystems cryostat. The sections were then incubated with primary antibodies against slow or fast myosin heavy chains (MyHC I (slow), 1:400, Sigma, USA, M8421 and MyHC II (fast), Sigma, USA, 1:400, M4276) and examined using a Leica Q500MC fluorescence microscope with an integrated digital camera (TCM 300F, Leica, Germany), 20× magnification. Image analysis was performed using ImageJ 1.52a software. At least 150 fibers were analyzed in each muscle sample.

### 4.7. Western Blotting

To isolate the total protein fraction and assess its desmin, α-actinin-2, α-actinin-3, and telethonin content, the RIPA reagent kit («Santa Cruz», Dallas, TX, USA) was used. The samples were diluted in a 2X sample electrophoresis buffer (5.4 mM Tris-HCl (pH 6.8), 4%-Ds-Na, 20%-glycerin, 10%-2-mercaptoethanol, and 0.02%-bromophenol blue). Electrophoresis was performed in 10% separating PAGE. The primary antibodies were used as follows: desmin («abcam», ab8592, 1:1000, Cambridge, United Kingdom), GAPDH («ABM», G041, 1:10,000 Richmond, BC, Canada), α-actinin-2 («SantaCruz», sc-17829, 1:1000, Dallas, TX USA), α-actinin-3 («MERCK», MABT143, 1:1000, Rahway, NJ USA), telethonin («abcam», ab210773, 1:1000, Cambridge, United Kingdom). As secondary antibodies, goat anti-rabbit antibodies conjugated with horseradish peroxidase («SantaCruz», Dallas, TX, USA) were used at a dilution of 1:50,000. The blots were revealed by using the Clarity Western ECL Substrate (BioRad Laboratories, Hercules, CA, USA). Protein bands were analyzed by using a C-DiGit Blot Scanner (Lincoln, NE, USA). Western blot data were processed by using Image Studio Digits Ver4.0 software (Lincoln, NE, USA).

### 4.8. Statistcal Analysis

Since the normal distribution of the sample was not confirmed in all cases, the nonparametric Kruskal–Wallis test was used to compare the experimental groups with each other. The data on protein content, active, and passive tension of isolated muscle were presented as mean ± standard error of the mean. The data on mRNA content were presented as median and interquartile range (0.25–0.75) ± minimum and maximum values.

## 5. Conclusions

In summary, in this study we showed that movement restriction leads to profound specific changes in the mechanical properties of skeletal muscles. The changes in mechanical properties are associated with the changes in the content of cytoskeletal proteins and collagen. One of the most noticeable results of movement restriction was a significant increase in the passive stiffness of soleus on the twenty-first day of the experiment, accompanied by a reduction in the same parameter in EDL. The obtained data on the state of the cytoskeleton did not allow us to give an exact explanation for this phenomenon. Future studies may include evaluation of signaling mechanisms regulating the synthesis, degradation, and mRNA expression of ECM and scaffold proteins of myofibers. However, given the significant involvement of soleus muscle in locomotor activity, for which high stiffness values may be excessive for successful performance, it can be assumed that limiting this activity may lead to an increase in the intrinsic stiffness of soleus.

## Figures and Tables

**Figure 1 ijms-24-04141-f001:**
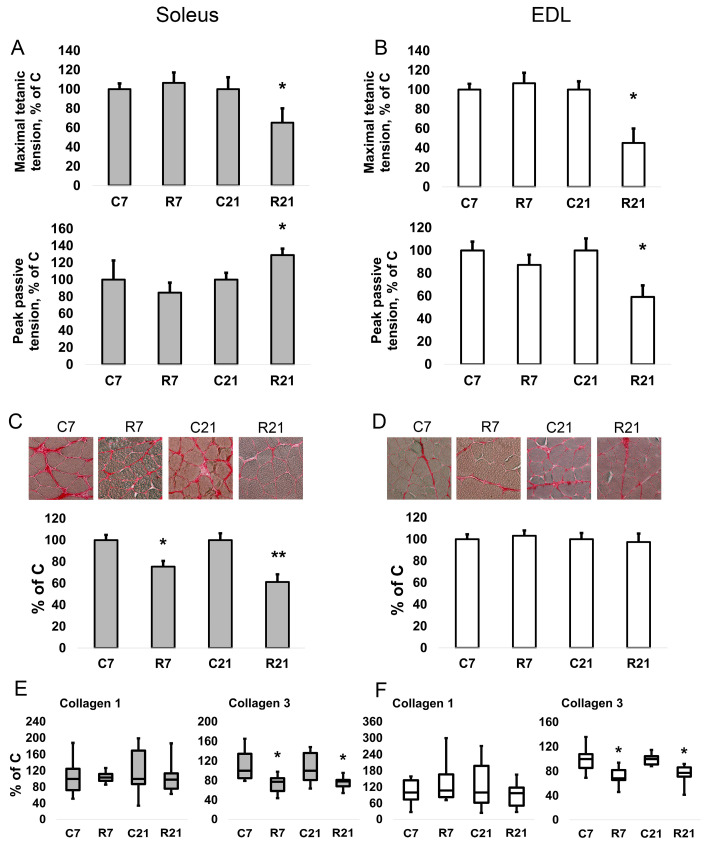
Maximal isometric tetanic tension and peak passive tension in soleus (**A**). Maximal isometric tetanic tension and peak passive tension in EDL (**B**). Histochemical detection of collagen in soleus (**C**). Histochemical detection of collagen in EDL (**D**). mRNA expression of collagen 1 and collagen 3 in soleus (**E**). mRNA expression of collagen 1 and collagen 3 in EDL (**F**). Data are shown as % of control group (Mean ± SEM); mRNA expression data are shown as median and interquartile range (0.25–0.75) ± the minimum and the maximum, *n* = 8 per group. *—significant difference from C, *p* < 0.05; **—significant difference from C, *p* < 0.001. C7 and C21—vivarium control for 7 and 21 days; R7 and R21—7-day and 21-day restriction of activity.

**Figure 2 ijms-24-04141-f002:**
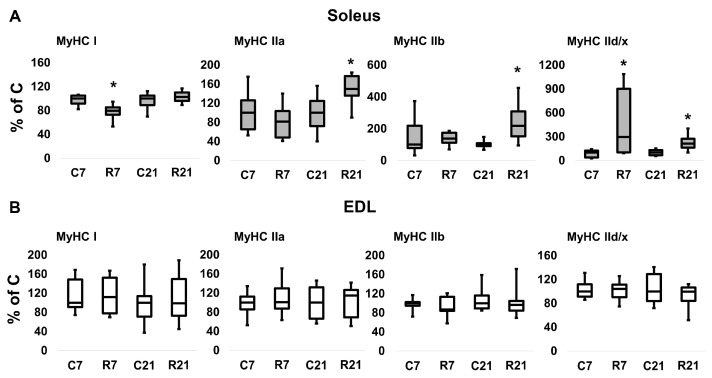
mRNA expression of MyHC I, MyHC IIa, MyHC IIb, MyHC IId/x in soleus (**A**) and EDL (**B**). Data are shown as % of control group (median and interquartile range (0.25–0.75) ± the minimum and the maximum), *n* = 8 per group. *—significant difference from C, *p* < 0.05. C7 and C21—vivarium control for 7 and 21 days; R7 and R21—7-day and 21-day restriction of activity.

**Figure 3 ijms-24-04141-f003:**
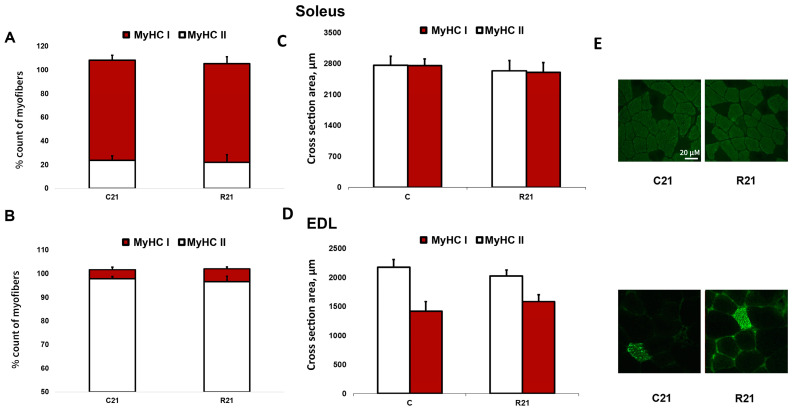
Immunohistochemical analysis of slow- (MyHC I) and fast-twitch (MyHC II) fiber number in cross sections of soleus (**A**) and EDL (**B**), cross-sectional area of soleus (**C**) and EDL (**D**) and representative immunohistochemical images (**E**). C21—vivarium control for 21 days; R21—21-day restriction of activity.

**Figure 4 ijms-24-04141-f004:**
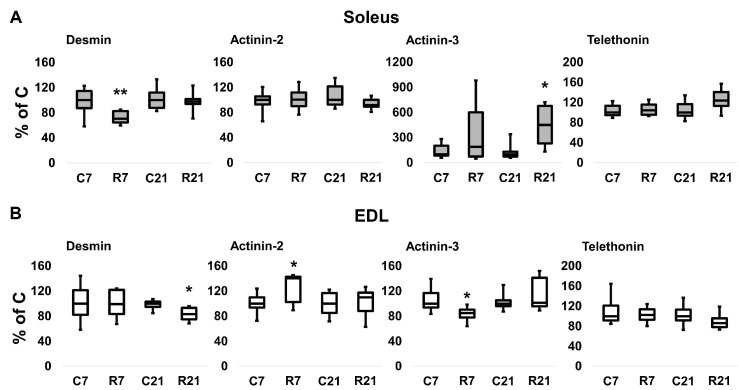
mRNA expression of desmin, α-actinin-2, α-actinin-3, and telethonin in soleus (**A**) and EDL (**B**). Data are shown as % of control group (median and interquartile range (0.25–0.75) ± the minimum and the maximum), *n* = 8 per group. *—significant difference from, *p* < 0.05; **—significant difference from C, *p* < 0.001. C7 and C21—vivarium control for 7 and 21 days; R7 and R21—7-day and 21-day restriction of activity.

**Figure 5 ijms-24-04141-f005:**
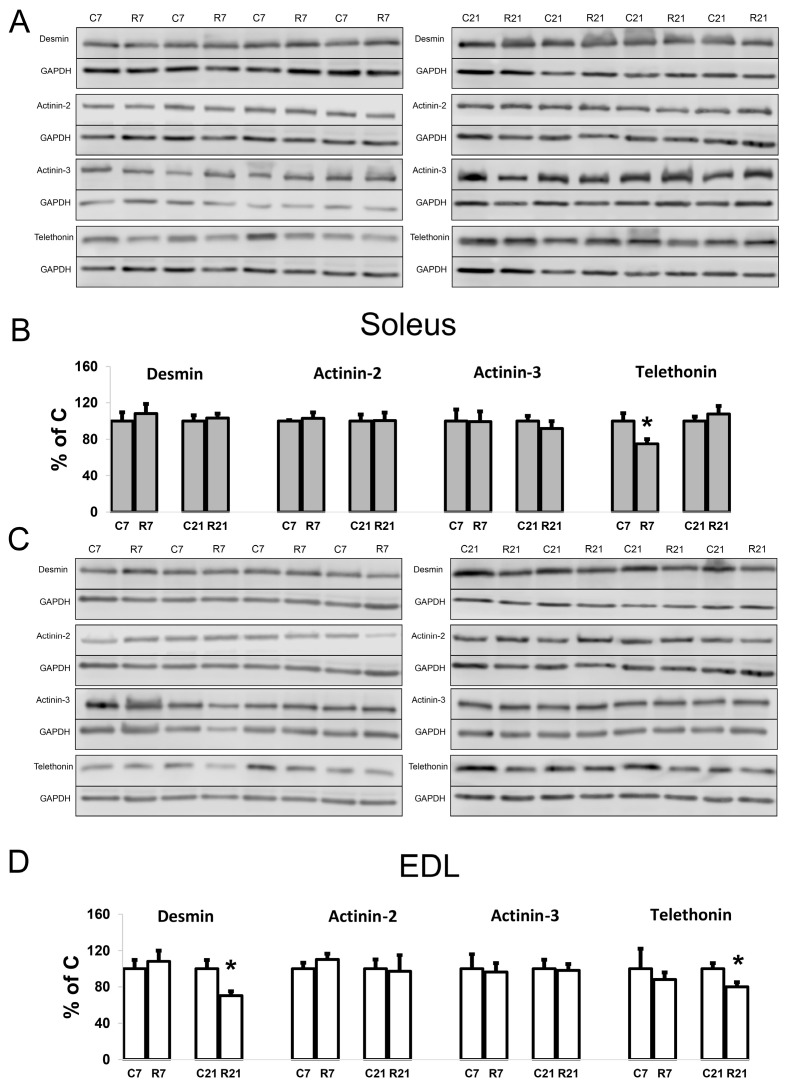
Representative immunoblots for the studied proteins in soleus (**A**). Quantification of desmin/GAPDH, α-actinin-2/GAPDH, α-actinin-3/GAPDH, and telethonin/GAPDH in soleus (**B**). Representative immunoblots for the studied proteins in EDL (**C**). Quantification of desmin/GAPDH, α-actinin-2/GAPDH, α-actinin-3/GAPDH, and telethonin/GAPDH in EDL (**D**). Data are shown as % of control group (Mean ± SEM), *n* = 8 per group. *—significant difference from C, *p* < 0.05. C7 and C21—vivarium control for 7 and 21 days; R7 and R21—7-day and 21-day restriction of activity.

**Figure 6 ijms-24-04141-f006:**
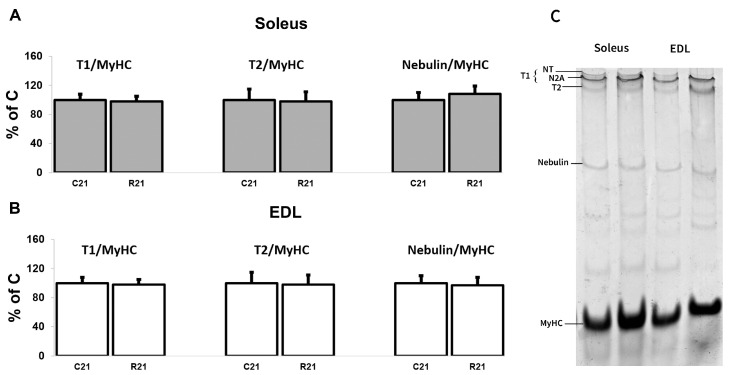
Quantification of T1/MyHC, T2/MyHC and nebulin/MyHC in soleus (**A**). Quantification of T1/MyHC, T2/MyHC, and nebulin/MyHC in EDL (**B**). Representative Coomassie Brilliant Blue stain for the studied proteins in soleus and EDL (**C**). Data are shown as % of control group (Mean ± SEM), *n* = 8 per group. C21—vivarium control for 21 days; R21—21-day restriction of activity.

**Figure 7 ijms-24-04141-f007:**
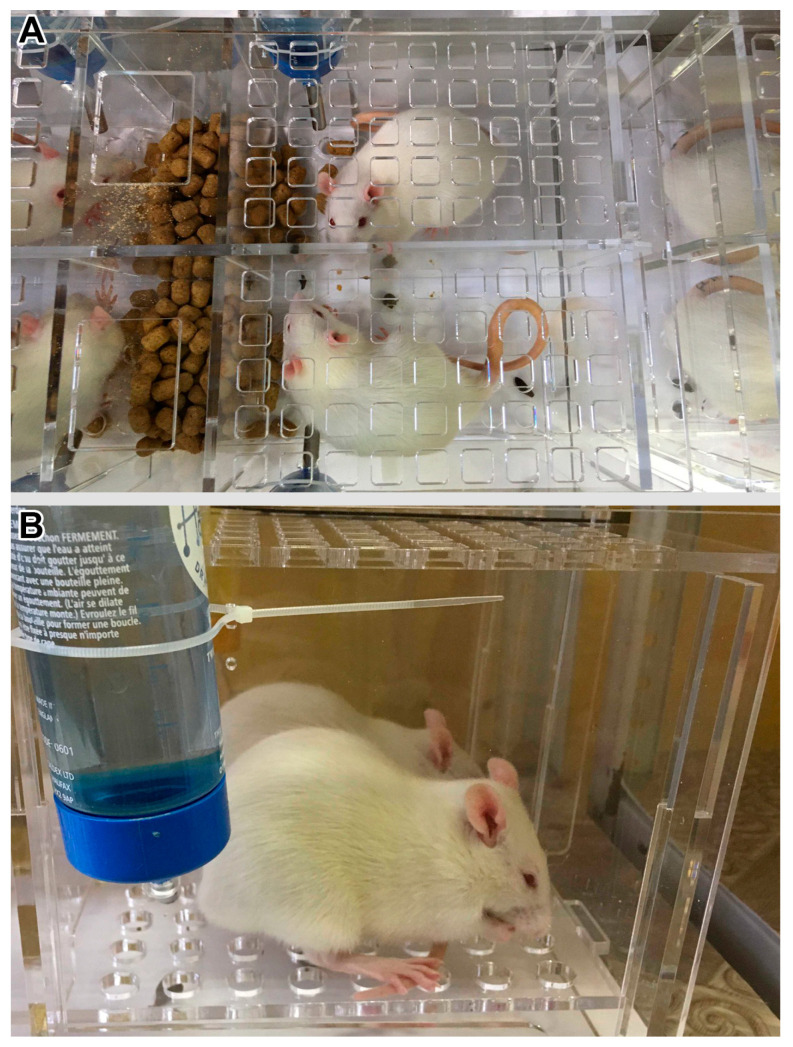
Design of the plexiglass restricted activity cages. View from the top (**A**) and from the side (**B**).

**Figure 8 ijms-24-04141-f008:**
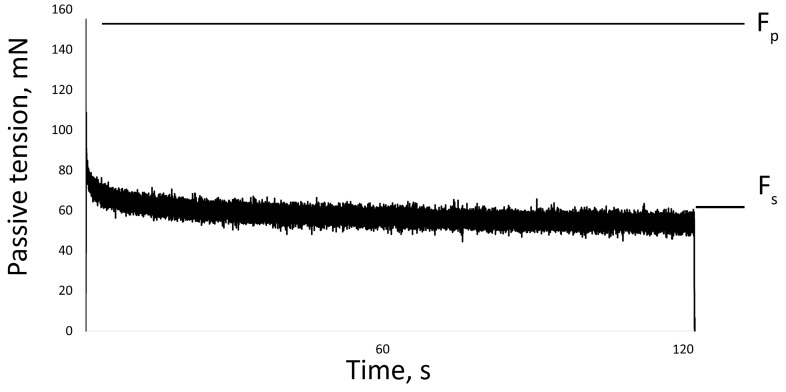
One recording of muscle stretch. Fp—peak tension, Fs—steady tension.

**Table 1 ijms-24-04141-t001:** Primer sequences.

Gene	Sequence
*α-actinin-2*	5′-CCGGGACTATCGTCGTAAGC-3′
	5′-CCAGCAATGTCCGACACCAT-3′
*α-actinin-3*	5′-ACTTTGACCGGAAGCGGAAT
	5′-ACCATGGTCATGATCCGAGC-3′
*desmin*	5′-TCTCAAGGGCACCAACGAC-3′
	5′-GGGTGTGACATCCGAGAGTG-3′
*Tcap (telethonin)*	5′-CCTTCTGGGCTGAGTGGAAA-3′
	5′-CTGCCGGTGGTAGGTCTCAT-3′
*collagen 1*	5′-ATCAGCCCAAACCCCAAGGAGA-3′
	5′-CGCAGGAAGGTCAGCTGGATAG-3′
*collagen 3*	5′-TGATGGGATCCAATGAGGGAGA-3′
	5′-GAGTCTCATGGCCTTGCGTGTTT-3′
*Myh7* (MyHC I)	5′-ACAGAGGAAGACAGGAAGAACCTAC-3′
	5′-GGGCTTCACAGGCATCCTTAG-3′
*Myh2* (MyHC IIa)	5′-TATCCTCAGGCTTCAAGATTTG-3′
	5′-TAAATAGAATCACATGGGGACA-3′
*Myh4* (MyHC IIb)	5′-CTGAGGAACAATCCAACGTC-3′
	5′-TTGTGTGATTTCTTCTGTCACCT-3′
*Myh1* (MyHC IId/x)	5′-CGCGAGGTTCACACCAAA-3′
	5′-TCCCAAAGTCGTAAGTACAAAATGG-3′

## Data Availability

The data presented in the study are available upon reasonable request from the corresponding author.

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
