# Peer review of "Changes in the Mechanical Properties of Fast and Slow Skeletal Muscle after 7 and 21 Days of Restricted Activity in Rats"

_ijms, 2023, doi:10.3390/ijms24044141_

Round 1

Reviewer 1 Report

The manuscript “Changes in mechanical properties of fast and slow skeletal muscle after 7- and 21-day restricted activity of rat” by S.A. Tyganov and coworkers deals with an interesting and less studied model of muscle disuse, i.e. the reduction of locomotor activity. This model is less severe than immobilization or unloading, as can be obtained with hindlimb suspension. Moreover, it is closer to the sedentary life style widely practised by healthy human beings. The authors have worked and published in the last decade on the model of hindlimb suspension, thus they are in the conditions to directly compare the two models. The few published studies on the model of restricted activity have not analysed in detail the functional properties of skeletal muscles at rest and during contraction. Moreover, they did not investigate the adaptation of the cytoskeleton molecular components. The novelty of the present study is the analysis of EDL and soleus ex vivo, representative of postural and locomotor muscles, respectively. The changes in mechanical properties at rest and in contraction are identified and possible molecular determinants of those changes are proposed. There are, however. critical points which I believe can be handled by the authors.

The paper is well written and easy to read, I would suggest to make the Introduction and the Discussion more concise. In contrast, the Results might become more reader-friendly with some additional information

I list here below some points which could be improved by the authors:

--page 2-3 (lines 89-107): first paragraph of Results: I think it would be very helpful to the readers if the authors could separate clearly: i) the sentence about rat and muscle mass, ii) the statements about contractile parameters and iii) the statement about passive mechanical properties: in the present version, there is no explanation that Fp is determined at rest, no explanation about E1 and E2 are. Since the Methods are at the end of the manuscript, the readers cannot imagine that we move from tetanic contraction to visco-elastic properties at rest.

Page 8, Discussion, line 191-193: The comparison with hind limb suspension is important, but the present study is about movement restriction. Thus, I would suggest to avoid starting the Discussion with hind limb suspension.

Page 8 line 202: change to “In that work….”

Page 8 line 209: would it be possible to compare the different studies on restricted mobility on the base of the size (squared cm) of the cage ?

Page 8 line 211: I would not use the word “immobility”, these experiments consider movement restriction

Page 10 line 286: the statement about “efficient operation of cytoskeleton..” is not strictly necessary. Thus, it may be removed. I would suggest to make the Discussion more concise and more focused on the message the authors want to give to the readers

Page 13 line 396: some more info about what are E1 and E2 should be given.

Author Response

The manuscript “Changes in mechanical properties of fast and slow skeletal muscle after 7- and 21-day restricted activity of rat” by S.A. Tyganov and coworkers deals with an interesting and less studied model of muscle disuse, i.e. the reduction of locomotor activity. This model is less severe than immobilization or unloading, as can be obtained with hindlimb suspension. Moreover, it is closer to the sedentary life style widely practised by healthy human beings. The authors have worked and published in the last decade on the model of hindlimb suspension, thus they are in the conditions to directly compare the two models. The few published studies on the model of restricted activity have not analysed in detail the functional properties of skeletal muscles at rest and during contraction. Moreover, they did not investigate the adaptation of the cytoskeleton molecular components. The novelty of the present study is the analysis of EDL and soleus ex vivo, representative of postural and locomotor muscles, respectively. The changes in mechanical properties at rest and in contraction are identified and possible molecular determinants of those changes are proposed. There are, however. critical points which I believe can be handled by the authors.

The paper is well written and easy to read, I would suggest to make the Introduction and the Discussion more concise. In contrast, the Results might become more reader-friendly with some additional information

We appreciate the Reviewer’s comments. We agree with the Reviewer about the importance to make the Introduction and the Discussion more concise. The comments seemed to be fair and reasonable, so we paid heed to the advice and suggestions, and the manuscript has been revised. We believe that the contents and the clarity of our paper are much improved in the revised version.

I list here below some points which could be improved by the authors:

--page 2-3 (lines 89-107): first paragraph of Results: I think it would be very helpful to the readers if the authors could separate clearly: i) the sentence about rat and muscle mass, ii) the statements about contractile parameters and iii) the statement about passive mechanical properties: in the present version, there is no explanation that Fp is determined at rest, no explanation about E1 and E2 are. Since the Methods are at the end of the manuscript, the readers cannot imagine that we move from tetanic contraction to viscoelastic properties at rest.

We totally agree with the comment. Corresponding correction has been made to the text of the Results section.

Page 8, Discussion, line 191-193: The comparison with hind limb suspension is important, but the present study is about movement restriction. Thus, I would suggest to avoid starting the Discussion with hind limb suspension.

Thank you for your comment. The Discussion section have been corrected.

Page 8 line 202: change to “In that work….”

Thank you for your comment. This line has been changed.

Page 8 line 209: would it be possible to compare the different studies on restricted mobility on the base of the size (squared cm) of the cage?

The discussion of the manuscript has been expanded.

Page 8 line 211: I would not use the word “immobility”, these experiments consider movement restriction

The word “immobility” has been changed to “movement restriction”.

Page 10 line 286: the statement about “efficient operation of cytoskeleton..” is not strictly necessary. Thus, it may be removed. I would suggest to make the Discussion more concise and more focused on the message the authors want to give to the readers

The aforementioned statement has been removed.

Page 13 line 396: some more info about what are E1 and E2 should be given.

Thank you for your comment. The information about E1 and E2 have been added to the Methods section in the Page 13.

With kind regards,

the authors of the manuscript.

Reviewer 2 Report

The manuscript is interesting and well written, however I have a few comments.

1.     Why were EDL and soleus muscle used for this experiment? These muscles are functionally very different. Soleus is a weight bearing muscle and is less influenced by decreased physical activity than EDL. It would be better to compare for example soleus and plantaris or gastrocnemius.

2.     Please provide better quality images of histochemical detection of collagen and if possible cross sections of whole muscles. It seems a bit unlikely that the content of collagen would decrease in 7 days to amount detectable by histochemical staining and image analysis. Did you also check if cross section/diameter of muscle fibres differed between groups?

3.     Why was immuinohistochemical analysis of all muscle fibre types not performed? Since MyHC II contains three different fibre types that have totally different properties it is not proper to regard them just as one group. Morphometric parameters of muscles of each fibre type would also be good addition.

4.     The quality of SDS-PAGE (Figure 6) and blots (Figure 4) seems not optimal, possibly influencing the results and lowering the sensitivity of the test.

5.     Please add a first paragraph to discussion section summarizing the findings of this study.

6.     First sentence of conclusion: you cannot generalise your results to all fast and slow skeletal muscles.

Author Response

The manuscript is interesting and well written, however I have a few comments.

We sincerely appreciate the effort and time devoted by the Reviewer while evaluating our manuscript. We have tried to address all the comments and make all necessary corrections in the manuscript.

  1. Why were EDL and soleus muscle used for this experiment? These muscles are functionally very different. Soleus is a weight bearing muscle and is less influenced by decreased physical activity than EDL. It would be better to compare for example soleus and plantaris or gastrocnemius.

We chose soleus and EDL as they are indeed very different in function and myosin phenotype. It was important for us to test the hypothesis that the movement restriction affects the locomotor muscles, and does not affect the postural muscles (because this model does not eliminate the weight bearing by soleus). Daily EMG durations are highest for soleus (11–15·h), intermediate for MG (5–9·h) and VL (3–14·h) and lowest for TA (2–3·h) (Hodgson, 2005). For EDL, the daily activity is even lower than for TA, as in normal rats, plantar flexion elicits a short phasic burst of EMG activity in EDL muscle, at the end of this movement (Sławińska, 1998). And, based on this, the comparison of soleus with either TA or EDL seemed most interesting.

We also chose EDL for it’s convenient tendons and similar size/length compared to soleus for use in ex vivo mechanical experiments.

  1. Please provide better quality images of histochemical detection of collagen and if possible cross sections of whole muscles. It seems a bit unlikely that the content of collagen would decrease in 7 days to amount detectable by histochemical staining and image analysis. Did you also check if cross section/diameter of muscle fibres differed between groups?

Thank you for your comment. We added the larger histochemical images of collagen staining to the supplementary file. We also added the information about muscle fibers CSA to the Figure 4.

  1. Why was immuinohistochemical analysis of all muscle fibre types not performed? Since MyHC II contains three different fibre types that have totally different properties it is not proper to regard them just as one group. Morphometric parameters of muscles of each fibre type would also be good addition.

The evaluation of the patterns of fast MyHC isoforms’ expression was not a subject of this study, as we were mainly focused on the effects of movement restriction on mechanical characteristics and cytoskeleton of EDL and soleus. Наши предварительные данные иммуногитологического окрашивания различных типов быстрых волокон не указывали на изменение миозинового фенотипа. Three “fast” MyHC isoforms have relatively close force parameters as compared to type I MyHC, thus, according to the current goals of the study we do not see the necessity for analyzing every MyHC isoform.

  1. The quality of SDS-PAGE (Figure 6) and blots (Figure 4) seems not optimal, possibly influencing the results and lowering the sensitivity of the test.

We check the quality of western blotting with Ponceau staining. We attach here оne representative image of the Ponceau staining done for this experiment:

Further image quality depends on the quality of antibodies specific to this protein, and we have done our best to achieve the best result. We have redone some representative blots in Figure 5.

  1. Please add a first paragraph to discussion section summarizing the findings of this study.

We have added the paragraph summarizing all the key findings of this work in the beginning of Discussion.

  1. First sentence of conclusion: you cannot generalise your results to all fast and slow skeletal muscles.

We totally agree with the comment. Corresponding correction has been made to the text of the Conclusion.

With kind regards,

the authors of the manuscript.

Round 2

Reviewer 2 Report

The authors satisfactorily addressed my comments.